

# Husbands' knowledge and attitudes regarding postpartum depression

Aisha M. Aqeeli[1,2], Hanan A. Badr[2] and Salmah A. Alghamdi[2]

[1] Labor and Delivery Department, East Jeddah Hospital, Jeddah, Saudi Arabia
[2] Maternity and Child Nursing Department, Faculty of Nursing, King Abdulaziz University, Jeddah, Saudi Arabia

## ABSTRACT

**Background.** Postpartum depression (PPD) is a prevalent mental health disorder that can occur anytime within the first year after childbirth. PPD has negative health consequences for mothers, infants, and other family members. Early detection and treatment are essential in mitigating these effects. This study aims to assess husbands' knowledge and attitudes toward PPD among men aged 20 years and older residing in Saudi Arabia.

**Methods.** This study employed a cross-sectional design. Participants were recruited through social media and face-to-face methods using a convenience sampling approach. A total of 401 husbands were included in the study. The data were analyzed using descriptive statistics and Pearson's simple correlation analysis.

**Results.** In this study, the majority of participants (89.5%, $n = 359$) were Saudi nationals. Approximately half of the husbands (48.4%, $n = 194$) were between the ages of 30 and 39, whereas only 10% ($n = 40$) were 50 or older. Nearly half of the participants (45.4%) demonstrated a high level of knowledge about PPD. Additionally, 66.1% of the husbands had a positive attitude toward PPD and had received prior information about the condition. Family and friends were the most commonly cited sources of PPD knowledge. A positive correlation was found between husbands' knowledge and attitudes toward PPD ($r = 0.117$, $P < 0.005$). Furthermore, significant associations were observed between husband's knowledge and attitude and several sociodemographic characteristics, including nationality, educational level, occupation, monthly income, and years of marriage ($P < 0.05$).

**Conclusion.** The husbands demonstrated a good level of knowledge and a positive attitude toward PPD. However, further research is needed to enhance their understanding and attitudes, particularly in addressing the negative beliefs about PPD identified in this study.

Corresponding author
Aisha M. Aqeeli, amaama14@hotmail.com

## INTRODUCTION

Postpartum depression (PPD) is defined as a depressive episode of moderate to severe intensity that begins 4 weeks after delivery and may persist for up to 12 months following childbirth (*Beck, 2002*). PDD is characterized by tearfulness, anxiety, emotional distress, irritability, sleep disorders, memory problems, guilt, loss of appetite, suicidal thoughts, feelings of weakness, and an inability to care for the baby (*American Psychiatric Association,*

*2013*). Unlike postpartum blues, which typically begin 3–4 days after childbirth and resolve by day 10 without treatment, PPD is not self-limiting and requires medical intervention (*Callahan & Caughey, 2018*).

PPD is a significant public health concern that affects the health outcomes of both mothers and infants, particularly in the Middle East (*Alshikh Ahmad, Alkhatib & Luo, 2021*; *Bintari, Sudibyo & Karimah, 2021*). The incidence of PPD has been rising among Arab women and is expected to affect at least one in five mothers (*Ayoub, Shaheen & Hajat, 2020*). The prevalence of PPD varies throughout the first postpartum year, with estimates ranging from 19% to 31% (*Alshikh Ahmad, Alkhatib & Luo, 2021*). In the Middle East, a systemic review by *Alshikh Ahmad, Alkhatib & Luo (2021)* reported a PPD prevalence of 27%. Economic hardships and pregnancy complications are among the most significant risk factors for PPD in these countries.

A recent multinational study conducted in Egypt, Yemen, Iraq, India, Ghana, and Syria found an overall prevalence of 13.6%. The lowest prevalence was recorded in Syria (2.3%), whereas the highest was in Ghana (26%) (*Amer et al., 2024*). In Saudi Arabia, PPD prevalence estimates vary by region with reported rates of 38.50% in Riyadh, 20.9% in Jeddah, 32.8% in Alkharj, and 75.7% in Jazan (*Abdelmola et al., 2023*; *Al Nasr et al., 2020*; *Alsayed et al., 2021*; *Alzahrani et al., 2022*).

According to a recent meta-analysis, risk factors for PPD in the Middle East include socioeconomic disadvantage, pregnancy-related health issues, low educational attainment, unintended pregnancies, the role of being a housewife, insufficient familial support, and formula feeding (*Alshikh Ahmad, Alkhatib & Luo, 2021*).

Although the exact etiology of PPD remains unknown, its development is closely linked to biological, psychological, socioeconomic, and cultural factors. Physical and biological risk factors include a history of depression, obesity, and hormonal changes. Psychological risk factors include low self-esteem, poor adaptability, and lack of parental knowledge. Additionally, inadequate support from a spouse or family contributes to emotional and social stress. Cultural factors, such as the societal preference for male offspring, have also been associated with an increased risk of PPD (*Agrawal, Mehendale & Malhotra, 2022*; *Alshikh Ahmad, Alkhatib & Luo, 2021*; *Ayoub, Shaheen & Hajat, 2020*; *Zhao & Zhang, 2020*).

PPD has serious health consequences for mothers, infants, and other family members. It can lead to instability and disruption in the mother's relationship with her spouse and family as well as impairing her bond with the infant. PPD is also associated with attachment issues and negative child developmental outcomes (*Saharoy et al., 2023*; *Slomian et al., 2019*). Women with PPD are significantly more likely to experience future episodes of depression and other mental and physical health problems (*Abdollahi & Zarghami, 2018*; *Zakeri et al., 2022*). Additionally, untreated PPD has been linked to an increased risk of suicide (*Slomian et al., 2019*) with approximately 2.3% of mothers who experienced PPD in 2020 reporting suicidal ideation (*Tabb et al., 2020*).

Early detection of PPD can significantly improve outcomes for both the mother and the family (*Ricci, Kyle & Carman, 2021*). However, mothers and caregivers often dismiss PDD

symptoms as a natural consequence of childbirth. Only 20% of women who experience PPD symptoms report them to their health care providers (*Anokye et al., 2018*).

Support systems for postpartum mothers must be knowledgeable about PPD symptoms to recognize them early and to encourage affected women to seek professional treatment (*Branquinho, Canavarro & Fonseca, 2019*). A study conducted in Saudi Arabia found that most women with PPD lived in nuclear family households with their husbands (*Al-Ghamdi et al., 2019*).

According to *Puswati & Suci (2019)*, the husband provides the most important support for postpartum mothers. He plays a crucial role in helping his wife maintain her health during the postpartum period by expressing love, care, and interest. Similarly, *Rosa, Apriyanti & Astuti (2021)* found that spousal support helps postpartum mothers manage emotional changes, provide care for their infants, practice self-care, and maintain a nutritious diet. Additionally, a study by *Ristanti & Masita (2020)* found a positive correlation between health-seeking behavior and husband support. In Saudi Arabia, a lack of spousal support has been identified as a significant predictor of PPD (*Al Nasr et al., 2020*). When husbands fail to provide adequate support, mothers may feel neglected and depressed, leading to negative attitudes and unfavorable behaviors (*Puswati & Suci, 2019*).

Existing studies on PPD that include husbands are limited and suggest that many husbands lack knowledge about PPD, which may hinder early recognition and support. For instance, in Thailand, *Juntaruksa, Prapawichar & Kaewprom (2017)* explored the knowledge and attitudes of postpartum women's husbands and female relatives regarding PPD. The study found that a significant number of husbands held misconceptions about the causes and risk factors of PPD. Similarly, recent studies conducted in Portugal, Malaysia, and India examining PPD awareness among the general population and family members revealed that partners and husbands—who serve as a critical support network for postpartum women—demonstrated a lack of knowledge about PPD (*Alsabi et al., 2022*; *Branquinho, Canavarro & Fonseca, 2019*; *Poreddi et al., 2020*).

In Saudi Arabia, *Alkhawaja et al. (2023)* conducted a recent study in the Eastern region to examine husbands' knowledge and attitudes toward postpartum depression. This study was carried out concurrently with the current study and was published earlier, revealing that a significant number of husbands lacked awareness of PPD symptoms and their impact and held negative attitudes toward the condition.

Husbands with good knowledge of PPD play a crucial role in the early detection of depressive symptoms in providing the necessary support for their wives, which significantly improves maternal recovery and overall well-being (*Ristanti & Masita, 2020*). Additionally, knowledgeable husbands are more likely to encourage their wives to seek professional help (*Pebryatie et al., 2022*). The recommendation to seek professional help for PPD has been associated with greater knowledge of PPD, lower stigma, and more positive attitudes toward the condition (*Branquinho, Canavarro & Fonseca, 2019*).

Therefore, it is essential to explore husbands' knowledge of and attitudes toward PPD. Additionally, most research in Saudi Arabia has focused on the prevalence and predictors of PPD, whereas studies on husbands' awareness of the condition remain limited. Given the

significant prevalence of PPD and the lack of research on this topic, further investigation is necessary.

This study aimed to assess husbands' knowledge and attitudes regarding PPD. The findings can inform health service delivery for postpartum women and their husbands, promoting greater awareness, fostering positive attitudes toward PPD, and addressing common misconceptions.

# THE RESEARCH QUESTION

What is the level of husbands' knowledge and attitudes regarding postpartum depression?

## Research hypotheses

H1: Husbands have a good level of knowledge and a positive attitude toward postpartum depression.

H2: Husbands with good knowledge of PPD will have a positive attitude toward postpartum depression.

# MATERIALS AND METHODS

## Study design

A descriptive cross-sectional design was used to assess husbands' knowledge and attitudes toward PPD.

## Sampling

A nonprobability convenience and snowball sampling method were used to recruit 401 husbands who met the inclusion criteria. Participants were eligible if they were 20 years or older, married, had at least one child; spoke Arabic; resided in Saudi Arabia; and were willing to participate in the study. The exclusion criteria included non-Arabic speakers and single males.

## Recruitment and study setting

A convenience sample was recruited online through social media and face-to-face methods. The researcher created digital flyers containing the study's title, aims, target population, and a barcode and web link to access the survey on Google Forms. These flyers were posted on social media platforms, including WhatsApp, Snapchat, Facebook, and Twitter. Additionally, the researcher distributed the survey to family members, friends, and health care providers, requesting that they share it with husbands who met the eligibility criteria.

The available participants who had visited a governmental hospital and a primary healthcare center in Jeddah City and met the inclusion criteria were recruited face-to-face. The researcher clearly explained the study's objectives. An electronic questionnaire was provided to participants *via* a barcode, which was distributed personally in the male waiting areas of the hospital and primary health care center.

## Ethical considerations

Ethical approval was obtained from the Ethical Committee of the Faculty of Nursing at King Abdulaziz University (KAU) in Jeddah (NREC Serial No: Ref No 2M.53). Additionally,

approval was granted by the Ethical Committee of the Ministry of Health in Jeddah (IRB Log No: A01592). Permission was then obtained from East Jeddah General Hospital and Al Jamiah Healthcare Center.

The flyer included the study title, objectives, target population, and a barcode and web link to access the survey on Google Forms. Upon accessing the web link, participants were presented with a brief explanation of the study on the first page of the questionnaire. Anonymity, confidentiality, and the right to refuse participation or withdraw from the study at any time were assured.

## Data collection tools

The questionnaire was modified and used with written permission from the author *Poreddi et al. (2020)*. The researcher translated the modified questionnaire from English to Arabic and then back to English. A bilingual expert reviewed the translation to ensure accuracy. The finalized questionnaire was then converted into an online format using Google Forms. The questionnaire consisted of three sections: (1) sociodemographic characteristics, (2) knowledge about the causes and symptoms of PPD, and (3) attitudes toward PPD.

### Part I: Sociodemographic characteristics

This section consisted of 11 questions covering participants' age, nationality, region, level of education, occupation, monthly income, years of marriage, number of living children, age of the youngest child, sex of children, and satisfaction with the sex of the baby. The questionnaire included both closed-ended (yes/no) questions and category questions with response options ranging from two to five per question.

### Part II: Knowledge about causes and symptoms of PPD

A self-reported questionnaire was adopted from *Chongpanish et al. (2014)* and modified by *Poreddi et al. (2020)*. This section included 37 questions divided into two parts. The first consisted of three closed-ended (yes/no) questions assessing the participants' prior information and experience with PPD. One of these questions (Question 2) was a multiple-choice item allowing the participants to select more than one source of information about PPD. The second contained 33 items, with 17 measuring knowledge of the causes of PPD, including three reverse-coded items (statements 6, 10, and 15), and 16 assessing knowledge of PPD symptoms, including four reverse-coded items (statements 2, 4, 5, and 16). Negatively worded items were reverse-coded before analysis. Responses for this section were categorized as "yes," "no," or "I do not know." A "yes" response was awarded one point, whereas "no" and "I do not know" responses received zero points. The total knowledge score ranged from 0 to 33 with higher scores indicating greater knowledge of PPD. The scoring system was categorized as follows:

(a) High knowledge level = (23–33)
(b) Moderate knowledge level = (12–22)
(c) Low knowledge level = (0–11)

### Part III. Attitudes toward PPD

To assess husbands' attitudes toward PPD, the researcher used a questionnaire developed by *Chongpanish et al. (2014)* and modified by *Poreddi et al. (2020)*. The questionnaire

consisted of 19 statements, including 13 reverse-coded items (statements 1, 2, 3, 5, 6, 8, 9, 10, 12, 15, 16, 18, and 19). Each item was presented in a three-category Likert scale format, with response options 3 (agree), 2 (partly agree), and 1 (disagree). The total attitude score ranged from 19 to 57 with higher scores indicating more positive attitudes toward PPD, whereas scores closer to 19 reflected more negative attitudes. Attitudes were categorized as positive if they exceeded the overall mean score and negative if they fell below the mean score.

## Validity and reliability

The Arabic version of the knowledge and attitude questionnaire was reviewed by five experts in nursing research at KAU. Three experts specialized in maternal and child nursing, whereas two specialized in psychiatric nursing. They assessed the content validity, completeness, and clarity of the items. Comments and suggestions were considered, and necessary modifications were made accordingly.

A pilot study was conducted online by distributing the Arabic version of the questionnaire *via* WhatsApp to 10% of the sample population (38 husbands) who met the inclusion criteria. Participants were asked to complete the questionnaire and provide feedback on any difficulties they encountered. Based on the pilot study results, no modifications were required. The 38 participants from the pilot study were included in the final study sample. Cronbach's alpha was calculated to assess the reliability of the study instruments. The total knowledge scale for PPD yielded a Cronbach's alpha of 0.859, indicating a high level of internal consistency. Similarly, the attitude scale had a Cronbach's alpha of 0.764, reflecting a good level of internal consistency.

## Data analysis

The data were analyzed using the Statistical Package for the Social Sciences (SPSS) program version 25 (IBM Corp., Armonk, NY, USA). Descriptive statistics, including frequency and percentage, were used to summarize sociodemographic characteristics. PPD knowledge and attitude scale variables were described using frequencies, percentages, mean scores, and standard deviations. The Kruskal–Wallis and Mann–Whitney U tests were conducted to examine the association between sociodemographic variables (categorical data) and husbands' knowledge and attitudes toward PPD. Pearson's simple correlation analysis was conducted to examine the relationship between PPD knowledge and attitude. A *P*-value of less than or equal to 0.05 was considered significant.

## RESULTS

A total of 652 participants responded to the survey. However, only 401 husbands who met the inclusion criteria were included in the study. The remaining 251 respondents were excluded, consisting of 203 females and 48 husbands who did not meet the eligibility criteria.

## Participants' sociodemographic characteristics

Table 1 presents the sociodemographic characteristics of the 401 husbands who participated in this study. The majority (89.5%, *n* = 359) were Saudi nationals. Nearly half (48.4%, *n*

= 194) were between the ages of 30 and 39, whereas 10% ($n = 40$) were 50 or older. Most participants (74.1%, $n = 297$) resided in the western region of Saudi Arabia. More than half (58.4%, $n = 234$) held bachelor's degrees, whereas only 1% ($n = 4$) and 1.7% ($n = 7$) had attained primary and intermediate education, respectively. Additionally, 64.8% ($n = 260$) worked in the government sector, whereas 2.7% ($n = 11$) were unemployed. Greater than half (54.1%, $n = 217$) had a monthly income exceeding 10,000 SAR per month.

Concerning marital duration and family composition, nearly three-quarters (70.3%, $n = 282$) had been married for more than 6 years. Additionally, 17.5% ($n = 70$) had been married for 4–6, whereas 12.2% ($n = 49$) had been married 1–3 years. Detailed sociodemographic data are presented in Table 1.

### Participants' responses to previous information and experience with PPD

Nearly two-thirds (63.1%, $n = 253$) of the participants reported that they had not encountered women with PPD. Additionally, a significant majority (84.5%, $n = 339$) stated that their wives had never been diagnosed with PPD.

### Participants' source of information about PPD

Among the 401 participants, nearly one-quarter (24.5%) received information from family and friends, whereas 21.4% relied on social media. Other sources of information included websites (19.2%), physicians or nurses (18.6%), miscellaneous sources (8.4%), and television (8%).

### Husbands' knowledge about causes of PPD

Table 2 presents the participants' responses regarding their knowledge of the causes and risk factors of PPD. Nearly two-thirds of the participants were aware of several contributing factors, including lack of family support (65.3%, $n = 262$), increased work pressure and stress (62.8%, $n = 252$), and poor relationships or marital conflicts (60.8%, $n = 244$). However, a significant proportion of the participants were unaware of other potential risk factors. For example, 84.0% ($n = 337$) did not recognize genetic or hereditary factors as a cause, 73.3% ($n = 294$) were unaware that poor maternal education could be a contributing factor, and 68.8% ($n = 276$) did not acknowledge disappointment with the sex of the baby as a possible risk factor. Additionally, around three-quarters of the respondents disagreed with misconceptions such as PPD being caused by possessed ghosts, sin, or black magic (76.3%, $n = 306$); being older than 20 years of age (73.6%, $n = 295$); and personal strength (78%, $n = 314$). The mean score of husbands' knowledge about the cause of PPD was ($M = 9.12$, $SD = 4.10$).

### Husbands' knowledge about symptoms of PPD

Table 3 presents the participant's responses regarding their knowledge of PPD symptoms. Most participants correctly identified common symptoms, including feeling sad or miserable (76.6%, $n = 307$), experiencing sleeping problems (76.6%, $n = 307$), feeling stressed or anxious (75.8%, $n = 304$), having crying spells for no reason (75.1%, $n = 301$), experiencing weight or appetite changes (71.3%, $n = 286$), and feeling irritable (70.8%, $n$

**Table 1  Socio-demographics characteristics of the participants ($n = 401$).**

| Socio-demographics characteristics | | Frequencies | % |
|---|---|---|---|
| **Nationality** | Saudi | 359 | 89.5% |
| | Non-Saudi | 42 | 10.5% |
| **Age** | 20 to <30 years | 58 | 14.5% |
| | 30 to <40 years | 194 | 48.4% |
| | 40 to <50 years | 109 | 27.2% |
| | ≥50 years | 40 | 10.0% |
| **Region** | Western region | 297 | 74.1% |
| | Southern region | 52 | 13.0% |
| | Central region | 26 | 6.5% |
| | Eastern region | 19 | 4.7% |
| | Northern region | 7 | 1.7% |
| **Education level** | Primary school | 4 | 1.0% |
| | Intermediate school | 7 | 1.7% |
| | High school | 89 | 22.2% |
| | Bachelor degree / Diploma | 234 | 58.4% |
| | Higher than Bachelor degree | 67 | 16.7% |
| **Occupation** | Government employee | 260 | 64.8% |
| | Private/business | 106 | 26.4% |
| | Free business | 24 | 6.0% |
| | Not working | 11 | 2.7% |
| **Monthly family income** | Less than 5,000 SAR (1,333.50 USD) | 50 | 12.5% |
| | 5,000 to 10,000 SAR (1,333.50 to 2,666.67 USD) | 134 | 33.4% |
| | More than 10,000 SAR (2,666.67 USD) | 217 | 54.1% |
| **Number of years of marriage** | 1 to 3 years | 49 | 12.2% |
| | 4 to 6 years | 70 | 17.5% |
| | More than 6 years | 282 | 70.3% |
| **Number of living children** | One | 79 | 19.7% |
| | Two | 111 | 27.7% |
| | Three | 78 | 19.5% |
| | Four | 55 | 13.7% |
| | Five and more | 78 | 19.5% |
| **Age of the youngest child** | 6 months and less | 60 | 15.0% |
| | 7 to 12 months | 58 | 14.5% |
| | 1 to 2 years | 73 | 18.2% |
| | More than 2 years | 210 | 52.4% |
| **Sex of the children** | Only males | 90 | 22.4% |
| | Only females | 78 | 19.5% |
| | Both males and females | 233 | 58.1% |
| **Are you satisfied with your children's sex** | No | 7 | 1.7% |
| | Yes | 394 | 98.3% |

= 284). Nearly two-thirds of respondents (64.8%, $n = 260$) disagreed that being interested in doing household tasks was a symptom of PPD, and three-quarters (75.8%, $n = 304$) disagreed that being interested in talking with others was indicative of PPD. Additionally,

**Table 2  Husbands' knowledge about causes of PPD (n = 401).**

| Causes | No/ Do not know | | Yes | | Mean | SD |
|---|---|---|---|---|---|---|
| | N | % | N | % | | |
| 1. Genetic/hereditary | 337 | 84.0% | 64 | 16.0% | .16 | .37 |
| 2. Crisis situation (Death of loved one, loss of job, divorce) in life | 224 | 55.9% | 177 | 44.1% | .44 | .50 |
| 3. Poor relationship/marital conflicts | 157 | 39.2% | 244 | 60.8% | .61 | .49 |
| 4. History of depression | 229 | 57.1% | 172 | 42.9% | .43 | .50 |
| 5. Lack of family support | 139 | 34.7% | 262 | 65.3% | .65 | .48 |
| 6. Ghost possessed or doing sin or Black magic* | 306 | 76.3% | 95 | 23.7% | .76 | .43 |
| 7. lack of confidence in taking care of baby | 173 | 43.1% | 228 | 56.9% | .57 | .50 |
| 8. Health problem/sickness of baby | 182 | 45.4% | 219 | 54.6% | .55 | .50 |
| 9. Domestic violence/husband violence | 180 | 44.9% | 221 | 55.1% | .55 | .50 |
| 10. Older age (more than 20 years)* | 295 | 73.6% | 106 | 26.4% | .74 | .44 |
| 11. Disappointment with sex of the baby | 276 | 68.8% | 125 | 31.2% | .31 | .46 |
| 12. Poverty/financial difficulties | 197 | 49.1% | 204 | 50.9% | .51 | .50 |
| 13. Poor education of the mother | 294 | 73.3% | 107 | 26.7% | .27 | .44 |
| 14. Increased work pressure/stress | 149 | 37.2% | 252 | 62.8% | .63 | .48 |
| 15. Personal strength* | 314 | 78.3% | 87 | 21.7% | .78 | .41 |
| 16. Substance abuse (Alcohol) among husband | 200 | 49.9% | 201 | 50.1% | .50 | .50 |
| 17. Single mother | 189 | 47.1% | 212 | 52.9% | .53 | .50 |

Notes.
*Reverse coded mean score: 9.12. SD: 4.10.

more than half of the participants (56.9%, $n = 228$) did not recognize a mother's bonding issues with her baby as a symptom of PPD. The mean score of husbands' knowledge about PPD symptoms was ($M = 11$, $SD = 3.74$).

Figure 1 illustrates husbands' overall knowledge levels regarding the causes and symptoms of PPD. Among the 401 participants, nearly half (45.4%, $n = 182$) demonstrated a high level of knowledge, whereas more than one-third (37.9%, $n = 152$) had a moderate level of knowledge. A smaller proportion (16.7%, $n = 67$) exhibited a low level of knowledge of PPD. The mean score for husbands' total knowledge of the causes and symptoms of PPD was ($M = 19.91$, $SD = 7.07$).

## The husbands' attitudes toward PPD

Table 4 presents the husbands' attitudes toward women with PPD. The majority of respondents (91.5%, $n = 367$) agreed that "we should be patient and empathetic toward women with PPD," followed by those who stated, "I am ready to help if a relative has PPD" (88.5%, $n = 355$). More than three-quarters of husbands disagreed with the statement: "PPD is an incurable disease and will continue to worsen" (85.3%, $n = 342$). Similarly, a significant proportion rejected the belief that "it is burdensome to take care of a woman who has PPD" (82.8%, $n = 332$) and that "mothers with PPD cannot be good mothers" (81.8%, $n = 328$). Additionally, the majority of husbands disagreed with the statements that "mothers with PPD should not have another child" (79.3%, $n = 318$), "should stay at home" (78.1%, $n = 313$), "are a burden to the family" (77.8%, $n = 312$), and "the
**Table 3  Husbands' knowledge about symptoms of PPD (n = 401).**

| Symptoms | No/ Do not know | | Yes | | Mean | Standard deviation SD |
|---|---|---|---|---|---|---|
| | N | % | N | % | | |
| 1. Feeling sad/miserable | 94 | 23.4% | 307 | 76.6% | .77 | .42 |
| 2. Mother bonding with baby* | 228 | 56.9% | 173 | 43.1% | .57 | .50 |
| 3. Worry about bonding with baby | 180 | 44.9% | 221 | 55.1% | .55 | .50 |
| 4. Interested in doing household tasks* | 260 | 64.8% | 141 | 35.2% | .65 | .48 |
| 5. Interested in talking with others* | 304 | 75.8% | 97 | 24.2% | .76 | .43 |
| 6. Feeling of fatigue/weakness | 128 | 31.9% | 273 | 68.1% | .68 | .47 |
| 7. Feeling stressed/anxious | 97 | 24.2% | 304 | 75.8% | .76 | .43 |
| 8. Loss of interest/pleasure | 127 | 31.7% | 274 | 68.3% | .68 | .47 |
| 9. Sleeping problems | 94 | 23.4% | 307 | 76.6% | .77 | .42 |
| 10. Lack of confidence | 136 | 33.9% | 265 | 66.1% | .66 | .47 |
| 11. Anger | 121 | 30.2% | 280 | 69.8% | .70 | .46 |
| 12. Weight/appetite changes | 115 | 28.7% | 286 | 71.3% | .71 | .45 |
| 13. Irritability | 117 | 29.2% | 284 | 70.8% | .71 | .46 |
| 14. Crying spells for no reason | 100 | 24.9% | 301 | 75.1% | .75 | .43 |
| 15. Death wishes | 177 | 44.1% | 224 | 55.9% | .56 | .50 |
| 16. Postpartum depression may occur after birth to 6 months only* | 260 | 64.8% | 141 | 35.2% | .65 | .48 |

**Notes.**
*Reverse coded mean score: 11. SD: 3.74.

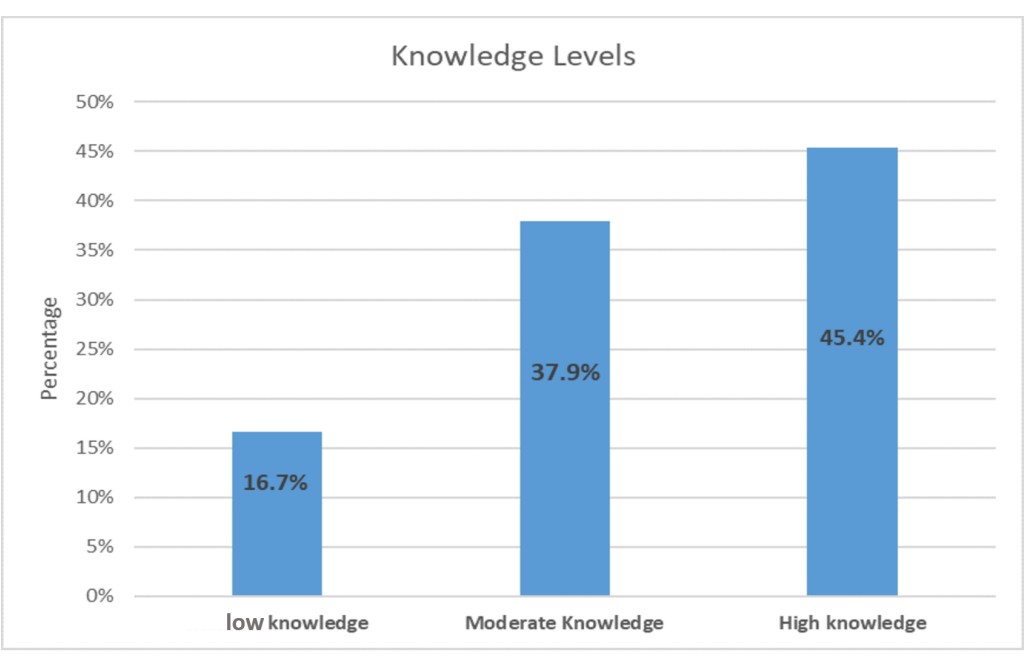

**Figure 1  The total knowledge about causes and symptoms of PPD.**

mother and baby should be separated" (70.1%, $n = 281$). Approximately two-thirds of husbands disagreed that "women with PPD do not require treatment" (67.8%, $n = 272$) and "they feel ashamed and do not tell anyone that a relative has PPD" (60.8%, $n = 244$). However, some negative attitudes toward women with PPD were observed. More than two-thirds of husbands (67.3%, $n = 270$) did not believe that PPD was common among women. Additionally, 59.1% ($n = 237$) disagreed with the notion that women with PPD are pitiable. Furthermore, 44.1% of the participants did not believe that postpartum depression increases the risk of suicide. Participants also demonstrated neutrality regarding certain views, such as whether "women with PPD should stop breast-feeding" (44.6%, $n = 179$) or "should be treated in a hospital" (40.9%, $n = 164$).

The total mean score on the attitude scale was ($M = 45.66$, $SD$, $3.5$). Based on the mean score (46), participants were classified as having a positive attitude ($>46$) or a negative attitude ($<46$). According to this classification, nearly two-thirds (66.1%) of the participants had a positive attitude toward women with PPD, whereas 33.9% had a negative attitude.

## Correlation between knowledge and attitudes toward PPD

Table 5 presents the correlation between husbands' knowledge and attitudes toward PPD. A significant positive correlation was found between the participants' knowledge and attitudes toward PPD ($r = 0.117$, $P < 0.019$), indicating that as husbands' knowledge about PPD increased, their attitudes became more positive.

## Relationship between knowledge and attitude mean score and sociodemographic characteristics of the husbands

Knowledge and attitude scores were not normally distributed among respondents. Therefore, the Mann–Whitney U-test and Kruskal-Wallis test were used to examine the relationship between sociodemographic variables (categorical data) and husbands' knowledge and attitudes about PPD with statistical significance set at ($P$-value $< 0.05$).

Table 6 presents the relationship between sociodemographic characteristics and husbands' mean knowledge scores. A significant association was found between the husband's knowledge and several sociodemographic factors, including nationality, educational level, occupation, monthly income, and years of marriage ($P < 0.05$). The results indicate that Saudi husbands had significantly higher knowledge scores about PPD than non-Saudi husbands ($P = 0.005$). Additionally, a significant correlation was observed between participants' education level and their knowledge ($P = 0.002$) with those holding a bachelor's degree, diploma, or higher demonstrating significantly greater knowledge compared to those with a primary or intermediate school education. Furthermore, a significant relationship was found between the husband's occupation and their knowledge scores ($P = 0.021$). Participants employed in government positions or self-employed businesses had significantly higher knowledge scores compared to those working in private businesses or those who were unemployed. The results also indicated a significant effect of income on knowledge scores ($P = 0.000$). Husbands with a high monthly income ($>10,000$ riyals) had significantly higher knowledge scores compared to those with lower monthly incomes ($<1,000$ and $< 5,000$ riyals). Additionally, a significant association was found

**Table 4  Husbands' attitudes towards PPD (n = 401).**

| Item | Agree (%) | | Partly Agree (%) | | Disagree (%) | | Mean | SD |
|---|---|---|---|---|---|---|---|---|
| | % | N | % | N | % | N | | |
| 1. I feel shame and do not tell anyone that my relative has postpartum depression[*] | 11.0% | 44 | 28.2% | 113 | 60.8% | 244 | 2.50 | .69 |
| 2. Postpartum women who have postpartum depression cannot be good mothers[*] | 6.7% | 27 | 11.5% | 46 | 81.8% | 328 | 2.75 | .57 |
| 3. Postpartum women who have postpartum depression should stay at home[*] | 8.2% | 33 | 13.7% | 55 | 78.1% | 313 | 2.70 | .61 |
| 4. We should be patient and have empathy with the women who have postpartum depression | 91.5% | 367 | 5.2% | 21 | 3.2% | 13 | 2.88 | .41 |
| 5. Postpartum women who have postpartum depression cannot take care of her own children[*] | 11.2% | 45 | 41.9% | 168 | 46.9% | 188 | 2.36 | .67 |
| 6. Postpartum depression is an uncured disease and will continue increasing in severity[*] | 6.2% | 25 | 8.5% | 34 | 85.3% | 342 | 2.79 | .54 |
| 7. Postpartum women who have postpartum depression are a pitiable person | 16.2% | 65 | 24.7% | 99 | 59.1% | 237 | 1.57 | .76 |
| 8. I feel it oppressive to take care of a woman who has postpartum depression[*] | 7.2% | 29 | 10.0% | 40 | 82.8% | 332 | 2.76 | .57 |
| 9. Postpartum women who have postpartum depression cannot make decisions at all[*] | 15.5% | 62 | 42.1% | 169 | 42.4% | 170 | 2.27 | .71 |
| 10. Postpartum women who have postpartum depression should not have another child[*] | 6.2% | 25 | 14.5% | 58 | 79.3% | 318 | 2.73 | .57 |
| 11. I am ready to help if my relative has postpartum depression | 88.5% | 355 | 6.7% | 27 | 4.7% | 19 | 2.84 | .48 |
| 12. Postpartum women who have postpartum depression is a burden to the family[*] | 7.0% | 28 | 15.2% | 61 | 77.8% | 312 | 2.71 | .59 |
| 13. Postpartum women who have postpartum depression should be treated in hospital | 25.2% | 101 | 40.9% | 164 | 33.9% | 136 | 1.91 | .76 |
| 14. Postpartum depression is an alert sign that a postpartum woman needs help from a caregiver | 51.1% | 205 | 37.2% | 149 | 11.7% | 47 | 2.39 | .69 |
| 15. If a woman develops postpartum depression, mother and baby to be separated[*] | 6.2% | 25 | 23.7% | 95 | 70.1% | 281 | 2.64 | .60 |
| 16. Postpartum depression does not require any treatment[*] | 5.7% | 23 | 26.4% | 106 | 67.8% | 272 | 2.62 | .59 |
| 17. Postpartum depression is common among women | 8.5% | 34 | 24.2% | 97 | 67.3% | 270 | 1.41 | .64 |
| 18. Breast feeding to be stopped if a woman develops postpartum depression[*] | 23.4% | 94 | 44.6% | 179 | 31.9% | 128 | 2.08 | .74 |
| 19. Woman with postpartum depression have a high risk of committing suicide[*] | 44.1% | 177 | 37.4% | 150 | 18.5% | 74 | 1.74 | .75 |

**Notes.**
[*]Reverse coded mean score: 45.66~46. SD: 3.50.

between the number of years of marriage and knowledge scores ($P = 0.046$). Participants who had been married for more than 6 years demonstrated significantly higher knowledge scores about PPD compared to those who had been married for less than 6 years. However, no significant association was found between PPD knowledge and other factors, including age, region, number of living children, age of the youngest child, sex of the children, and satisfaction with the sex of their children.

**Table 5** Correlation between knowledge and attitudes towards PPD ($n = 401$).

| | | Husbands' attitudes toward postpartum depression |
|---|---|---|
| Husbands' knowledge about postpartum depression | Pearson correlation | .117 |
| | Sig. (2-tailed) | .019[*] |

**Notes.**
[*]$P$-value < 0.05.

Table 7 presents the relationship between sociodemographic characteristics and husbands' attitude mean scores. The analysis showed no significant association between participants' attitudes toward PPD and sociodemographic variables, including nationality ($P = 0.393$), age ($P = 0.438$), region ($P = 0.976$), education level ($P = 0.058$), occupation ($P = 0.699$), monthly family income ($P = 0.069$) number of years of marriage ($P = 0.170$), number of living children ($P = 0.949$), age of the youngest child ($P = 0.928$), sex of the children ($P = 0.793$), and satisfaction with the sex of children ($P = 0.115$).

## DISCUSSION

This study explored husbands' knowledge and attitudes toward PPD in Saudi Arabia. The findings indicate that participants generally demonstrated a good level of knowledge and positive attitudes toward PPD. However, some misunderstandings and negative beliefs about PPD persist among certain participants. Additionally, the study revealed a significant positive correlation between husbands' knowledge and their attitudes toward PPD.

In this study, 45.4% of husbands exhibited a high level of knowledge about PPD. This finding suggests that a significant proportion of husbands are aware of PPD, which can play a crucial role in recognizing and addressing the condition while providing necessary support. However, it also highlights the need for further education and awareness. These results are consistent with previous studies indicating that family members, including relatives and husbands, have good knowledge of PPD (*Juntaruksa, Prapawichar & Kaewprom, 2017*; *Poreddi et al., 2020*). Additionally, the present study's findings align with recent studies conducted among the general population (*Alsabi et al., 2022*; *Branquinho, Canavarro & Fonseca, 2019*). However, these findings contrast with a study conducted in the eastern region of Saudi Arabia, which assessed husbands' knowledge and attitudes toward PPD and found that more than half of the participants had inadequate knowledge (*Alkhawaja et al., 2023*). Differences in these results are likely due to variations in socioeconomic status and cultural factors.

Regarding participants' knowledge of the causes and risk factors of PPD, this study found that nearly two-thirds (62.8%) were aware that a lack of family support and marital conflict are risk factors for PPD. This awareness is crucial because it can lead to proactive efforts to provide husbands with support and address relationship challenges during the postpartum period. Given the essential role of spousal support during this time, engaging husbands in both prenatal and postnatal care education is vital. Currently, husbands are permitted to attend follow-up visits and educational sessions. However, greater efforts should be made

**Table 6  Relationship between socio-demographic characteristics and knowledge mean score of husbands ($n = 401$).**

| Variables | N | Mean rank | Test statistic | P-value |
|---|---|---|---|---|
| **Nationality** | | | | |
| Saudi | 359 | 206.61 | 5524[a] | 0.005[*] |
| Non-Saudi | 42 | 153.02 | | |
| **Age** | | | | |
| 20 to <30 years | 58 | 166.43 | | |
| 30 to <40 years | 194 | 210.06 | 7.147[b] | 0.067 |
| 40 to <50 years | 109 | 207.63 | | |
| ≥50 years | 40 | 189.11 | | |
| **Region** | | | | |
| Western region | 297 | 201.62 | | |
| Southern region | 52 | 195.75 | | |
| Central region | 26 | 196.75 | 1.596[b] | 0.810 |
| Eastern region | 19 | 224.58 | | |
| Northern region | 7 | 165.57 | | |
| **Education level** | | | | |
| Primary school | 4 | 105.50 | | |
| Intermediate school | 7 | 112.43 | | |
| High school | 89 | 176.32 | 16.749[b] | 0.002[*] |
| Bachelor degree/Diploma | 234 | 205.11 | | |
| Higher than Bachelor degree | 67 | 234.39 | | |
| **Occupation** | | | | |
| Government employee | 260 | 206.98 | | |
| Private /business | 106 | 191.80 | 9.715[b] | 0.021[*] |
| Free business | 24 | 221.06 | | |
| Not working | 11 | 104.64 | | |
| **Monthly family income** | | | | |
| Less than 5,000 riyals | 50 | 141.61 | | |
| 5,000 to 10,000 riyals | 134 | 189.35 | 21.598[b] | 0.000[**] |
| More than 10,000 riyals | 217 | 221.88 | | |
| **Number of years of marriage** | | | | |
| 1 to 3 years | 49 | 169.66 | | |
| 4 to 6 years | 70 | 187.49 | 6.178[b] | 0.046[*] |
| More than 6 years | 282 | 209.80 | | |
| **Number of living children** | | | | |
| One child | 79 | 188.89 | | |
| Two children | 111 | 205.47 | | |
| Three children | 78 | 202.54 | 2.377[b] | 0.667 |
| Four children | 55 | 217.01 | | |
| Five and more children | 78 | 194.07 | | |

**Table 6** (*continued*)

| Variables | N | Mean rank | Test statistic | P-value |
|---|---|---|---|---|
| **Age of the youngest child** | | | | |
| 6 months and less | 60 | 174.05 | | |
| 7 to 12 months | 58 | 207.10 | 3.907[b] | 0.272 |
| 1 to 2 years | 73 | 202.39 | | |
| More than 2 years | 210 | 206.53 | | |
| **Sex of the children** | | | | |
| Only males | 90 | 185.38 | | |
| Only females | 78 | 192.43 | 3.448[b] | 0.187 |
| Both males and females | 233 | 209.90 | | |
| **Satisfaction with the sex** | | | | |
| No | 7 | 132.64 | | |
| Yes | 394 | 202.21 | 900.500 | 0.115 |

**Notes.**
[*]Significant at 0.05.
[**]significant at 0.01.
[a]Mann–Whitney U value.
[b]Kruskal–Walli's test value.

to encourage their participation in these sessions. PPD is a complex condition influenced by multiple factors; therefore, it is essential for husbands to be aware of these contributing factors and to provide necessary support. These findings align with previous studies that have also identified poor family relationships as a major risk factor for PPD (*Branquinho, Canavarro & Fonseca, 2019*; *Juntaruksa, Prapawichar & Kaewprom, 2017*; *Poreddi et al., 2020*). In contrast, a study by *Alkhawaja et al. (2023)* found that most husbands were unaware that poor education, young maternal age, and genetic predisposition are risk factors for PPD.

Regarding the signs and symptoms of PPD, this study found that most participants correctly identified its prevalent symptoms, indicating a general awareness of common PPD signs. These findings align with previous research (*Alsabi et al., 2022*; *Alkhawaja et al., 2023*; *Juntaruksa, Prapawichar & Kaewprom, 2017*; *Poreddi et al., 2020*) and highlight the importance of early detection and intervention. However, continued efforts to raise awareness about PPD remain crucial. Enhancing understanding and recognition of symptoms can facilitate prompt medical attention, ultimately leading to improved outcomes for mothers and their families.

The results of this study showed that two-thirds of the participants (66.1%) had a generally positive attitude toward PPD. This suggests a degree of empathy and understanding, which can be crucial in providing support to wives experiencing PPD. It also indicates that efforts to raise awareness and educate the public about PPD may have a positive impact. The findings of this study are consistent with previous research conducted among family members (*Juntaruksa, Prapawichar & Kaewprom, 2017*; *Poreddi et al., 2020*) and the general population (*Branquinho, Canavarro & Fonseca, 2019*). However, these results contrast with studies that found participants held negative attitudes toward PPD (*Alkhawaja et al., 2023*; *Alsabi et al., 2022*). This discrepancy may be attributed to misconceptions and stigma surrounding PPD.

**Table 7** Relationship between socio-demographic characteristics and attitude mean score of husband ($n = 401$).

| Variables | N | Mean rank | Test statistic | P-value |
|---|---|---|---|---|
| **Nationality** | | | | |
| Saudi | 359 | 199.32 | 6935[a] | 0.393 |
| Non-Saudi | 42 | 215.37 | | |
| **Age** | | | | |
| 20 to <30 years | 58 | 194.43 | | |
| 30 to <40 years | 194 | 196.30 | 2.714[b] | 0.438 |
| 40 to <50 years | 109 | 216.35 | | |
| ≥50 years | 40 | 191.51 | | |
| **Region** | | | | |
| Western region | 297 | 200.30 | | |
| Southern region | 52 | 207.75 | | |
| Central region | 26 | 196.62 | 0.476[b] | 0.976 |
| Eastern region | 19 | 193.05 | | |
| Northern region | 7 | 218.36 | | |
| **Education level** | | | | |
| Primary school | 4 | 127.50 | | |
| Intermediate school | 7 | 261.79 | | |
| High school | 89 | 186.36 | 9.124[b] | 0.058 |
| Bachelor degree/Diploma | 234 | 197.99 | | |
| Higher than Bachelor degree | 67 | 229.01 | | |
| **Occupation** | | | | |
| Government employee | 260 | 202.64 | | |
| Private/business | 106 | 198.37 | 1.427[b] | 0.699 |
| Free business | 24 | 181.92 | | |
| Not working | 11 | 229.27 | | |
| **Monthly family income** | | | | |
| Less than 5,000 riyals | 50 | 198.14 | | |
| From 5,000 to 10,000 riyals | 134 | 183.37 | 5.342[b] | 0.069 |
| More than 10,000 riyals | 217 | 212.55 | | |
| **Number of years of marriage** | | | | |
| 1 to 3 years | 49 | 191.64 | | |
| 4 to 6 years | 70 | 180.33 | 3.542[b] | 0.170 |
| More than 6 years | 282 | 207.76 | | |
| **Number of living children** | | | | |
| One child | 79 | 193.55 | | |
| Two children | 111 | 204.44 | | |
| Three children | 78 | 197.85 | 0.723[b] | 0.949 |
| Four children | 55 | 200.29 | | |
| Five and more | 78 | 207.31 | | |

**Table 7** (*continued*)

| Variables | N | Mean rank | Test statistic | P-value |
|---|---|---|---|---|
| **Age of the youngest child** | | | | |
| 6 months and less | 60 | 203.48 | | |
| 7 to 12 months | 58 | 191.64 | 0.459[b] | 0.928 |
| 1 to 2 years | 73 | 201.39 | | |
| More than 2 years | 210 | 202.74 | | |
| **Sex of the children** | | | | |
| Only males | 90 | 196.15 | | |
| Only females | 78 | 196.65 | 0.464[b] | 0.793 |
| Both males and females | 233 | 204.33 | | |
| **Satisfaction with the sex** | | | | |
| No | 7 | 65.00 | 900.500[a] | 0.115 |
| Yes | 394 | 203.42 | | |

**Notes.**
*Significant at 0.05.
**Significant at 0.01.
[a] Mann–Whitney U value.
[b] Kruskal–Walli's test value.

This study found a significant positive correlation between husbands' knowledge and attitudes toward PPD. This suggests that as husbands' knowledge about PPD increases, their attitudes become more positive. This finding has important implications for supporting families affected by PPD. When husbands are well-informed about the condition, they may exhibit greater empathy, understanding, and support toward their partners, ultimately leading to better outcomes for both mothers and the entire family unit. The findings of this study are supported by previous research conducted by *Poreddi et al. (2020)* and *Branquinho, Canavarro & Fonseca (2019)*, which similarly found that participants with higher knowledge scores demonstrated more positive attitudes toward PPD.

However, this finding contrasts with the study by *Alsabi et al. (2022)*, which reported a significant level of knowledge about PPD but a negative attitude and low awareness of PPD among social support networks for postnatal women. This discrepancy may be attributed to differences in the study population. *Alsabi et al.*'s (*2022*) study primarily included women who had limited exposure to public knowledge about PPD.

The results of this study indicated a significant relationship between husbands' knowledge mean scores about PPD and factors including nationality, level of education, occupation, monthly income, and the number of years of marriage ($P < 0.05$). However, no significant relationship was found with age, region, number of living children, age of the youngest child, gender of the children, or gender satisfaction. These findings provide valuable insights into the factors that may influence husbands' PPD knowledge levels. Additionally, these results align with the findings of *Alkhawaja et al. (2023)*, which also reported a significant association between husbands' knowledge about PPD and factors such as level of education and monthly income ($P < 0.05$).

Regarding the effect of education on husbands' knowledge about PPD, a significant difference was observed among educational levels. Husbands with a bachelor's degree or higher demonstrated greater knowledge compared to those with primary or intermediate

school education. The findings of this study are consistent with research by *Alkhawaja et al. (2023)*, *Poreddi et al. (2020)*, and *Branquinho, Canavarro & Fonseca (2019)*, which also found that individuals with higher education levels had greater knowledge about PPD than those with lower levels of education. The results suggest that more educated husbands are not only more knowledgeable about PPD but also more likely to understand effective treatment options and recognize its symptoms. In contrast, a study by *Alsabi et al. (2022)* reported no significant association between educational level and knowledge about PPD, highlighting potential differences in study populations or access to PPD-related information.

The findings of our study suggest that there was no significant relationship between husbands' attitude mean scores toward PPD and demographic factors such as nationality, education, occupation, monthly income, years of marriage, number of living children, age of the youngest child, and gender satisfaction. This indicates that these demographic factors did not notably influence husbands' attitudes toward PPD. It is important to acknowledge that these results may be specific to the sample and methodology used in this study. Similar findings were reported by *Alkhawaja et al. (2023)*, which also found no significant relationship between husbands' attitudes toward PPD and demographic factors.

Although our study found that demographic factors did not significantly influence husbands' attitudes toward PPD, previous research has suggested that certain demographic factors can affect attitudes toward PPD (*Alsabi et al., 2022*; *Branquinho, Canavarro & Fonseca, 2019*; *Poreddi et al., 2020*).

Our findings indicated no relationship between participants' attitudes toward PPD and their education levels. This contrasts with the study by *Branquinho, Canavarro & Fonseca (2019)*, which found that males and other social support networks with lower education levels held more negative attitudes toward PPD. Similarly, our results contradict those of *Poreddi et al. (2020)*, which reported that family members with higher education levels exhibited more positive attitudes toward PPD compared to those with lower education levels.

Furthermore, our study found no relationship between the number of children and participants' attitudes toward PPD. This contradicts the findings of a previous study that suggested that parents, including both wives and husbands with children, have a more favorable attitude toward PPD than those without children (*Alsabi et al., 2022*).

Regarding the overall results of the present study, H1, and H2 were accepted.

## STRENGTHS AND LIMITATIONS

This study is one of the first nursing studies conducted to explore husbands' knowledge and attitudes towards PPD in Saudi Arabia. A key strength of this research is the use of an electronic self-administered questionnaire, which facilitated the collection of a large amount of data through cost-effective methods. Multiple recruitment strategies were employed to reach a diverse and substantial number of participants. Additionally, the online survey format enhanced convenience, allowing husbands to respond at their own pace. However, this study has several limitations that should be acknowledged. First, the

cross-sectional design provides only a snapshot of knowledge and attitudes at a specific point in time, limiting the ability to establish causality or track changes over time. The study's focus on a specific group (husbands) limits the generalizability of the findings to a broader population. Additionally, the sample was predominantly composed of participants from the western region of Saudi Arabia with varying numbers of children and a high level of education, which may not fully represent husbands across the country. Another limitation was the relatively low response rate from social media recruitment, which may have affected the diversity of the sample. Finally, the limited availability of prior research on husbands' knowledge and attitudes regarding PPD posed a challenge in contextualizing the findings within the existing literature.

## CONCLUSION

The study findings demonstrated that husbands exhibited a good level of knowledge and generally positive attitudes toward PPD. However, there remains a need to enhance understanding and address misconceptions related to PPD among some participants. The results highlight the importance of developing strategies to improve PPD awareness in postnatal care and actively involving husbands in health education programs to help recognize the early signs and symptoms of PPD. Additionally, educational campaigns and targeted interventions are essential to further improve knowledge and promote positive attitudes about PPD among husbands and the general public.

### Funding
The authors received no funding for this work.

### Competing Interests
The authors declare there are no competing interests.

### Author Contributions
- Aisha M. Aqeeli conceived and designed the experiments, performed the experiments, analyzed the data, prepared figures and/or tables, authored or reviewed drafts of the article, and approved the final draft.
- Hanan A. Badr conceived and designed the experiments, performed the experiments, analyzed the data, prepared figures and/or tables, authored or reviewed drafts of the article, and approved the final draft.
- Salmah A. Alghamdi conceived and designed the experiments, performed the experiments, analyzed the data, prepared figures and/or tables, authored or reviewed drafts of the article, and approved the final draft.

### Human Ethics
The following information was supplied relating to ethical approvals (i.e., approving body and any reference numbers):

Ethical approval was obtained from the Ethical Committee of the Faculty of Nursing at King Abdul-Aziz University (KAU) in Jeddah (NREC Serial No: Ref No 2M. 53). In addition, ethical approval was obtained from the Ethical Committee of Ministry of Health in Jeddah (reference number IRB Log No: A01592). Then, permission was obtained from East Jeddah General Hospital and Al Jamiah Healthcare Center.

## Data Availability

The code is available in the Supplemental Files.

## Supplemental Information

Supplemental information for this article can be found online at http://dx.doi.org/10.7717/peerj.19426#supplemental-information.

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
