# Peer review of "Husbands’ knowledge and attitudes regarding postpartum depression"

_PeerJ, doi:10.7717/peerj.19426_

## Round 0.1 · original submission · Major Revisions

In addition to the reviewers' comments, the authors should adress the following ramarks:
- Of the main concern is that the authors included those who have not heard of PPD to calclclate the score of knowledge. Yet, the level should be calculated from those who heard of this condition before (n=265) (we can note evaluate the knowledge of one who have basically not heard of PPD).
-Extend the results of the abstract
- You should revise the flow of your ideas in the introduction. The definition (lines 54-57) should be provided before the epidemiology.
Provide the exclusion criteria in the methods.
Line 144: which author?
The discusion should be extensively summarised by discussing the most important results. The discussion of the demographic characteristics has no sense. In addition, it is better to remove the subtitles and avoid to repeat exaggeratedly the results.
At last, the manuscript should be edited for English language and the way of writing (you do not need to repeat the 33.1% have not heard of PPD if you stated that 66.9% knew it for example).

Reviewer 1 ·

Basic reporting

a. Line 43-44 doesn’t make sense given that postpartum depression can only occur after birth. You might mean that the incidence of depression is highest during the postpartum period?
b. The Shorey 2018 incidence of PPD is a little outdated, please find a more recent report.
c. Please add in some information about the cultural aspect of PPD specifically in the middle east as culture towards how new mothers should act and certain traditions play a role in PPD differently across the world. This would be important to address in the introduction
d. It’s hard to understand what the purpose of this paper is from the introduction, please reframe it to talk more about husband’s role and why it’s important to understand husband’s awareness of PPD.

Experimental design

a. This study does not add very much to the field and is not novel. Authors can benefit from explaining more in the introduction and discussion why these results are novel and fill a gap in the research.
b. The research question was not well defined and should have been added to the introduction as well.
c. Clarity as to why this research is being done in the specific area needs to be addressed.

Validity of the findings

a. Most of the results are just frequencies that do not tell the scientific community much. Though there is one correlation shown this is not enough. I would suggest for stronger statistical tests to be done to understand trends in knowledge and attitudes. Potentially look at regression models or a latent class analysis.

Additional comments

The large sample size of this study is a great strength yet the authors do not clearly define how this study adds to the academic community nor do they take advantage of the large sample size by reporting findings beyond just frequencies. I believe there is scope to improve this publication.

·

Basic reporting

1) At the abstract, and before the methods in the abstract, the objectives should be SMART (time, person, place, setting).
2) Add the age of husbands in the results of the abstract.
3)45 Globally, the incidence of PPD is 12%, whereas the general prevalence of depression is 17% among healthy mothers without any history of depression (Shorey et al., 2018). The Middle East has the highest prevalence of PPD (26%), whereas Europe has the lowest (8%; Shorey et al., 2018). ------- single reference  Kindly add the proper references for each.

4) Lines 45-53 contain information about the global prevalence of PPD up to 2021. You can use the prevalences from a recent multinational study in 2024 (DOI: 10.1186/s12889-024-18502-0).

5) Line 69: and newborns—only up to the first month (which can persist to the first year)


6) 515-516 ( This is the first nursing study of its kind conducted in Saudi Arabia to assess husbands' knowledge
516 and attitude toward PPD) add to the strength not a conclusion

7) he results highlight the importance of
519 developing strategies to improve awareness of PPD for postnatal care and including husbands in
520 health education to recognize early signs and symptoms of PPD. Additionally, there is a need for
521 educational campaigns and interventions to improve knowledge and promote positive attitudes
522 about PPD among husbands and the public.----- its not a conclusion it's a recommendation


8) rewrite the conclusion

9 ) Does these husbands have family history of PPD
what kind of support they provide

11) at the end of each table what is the meaning of negatively worded-----

12) in the introduction add more details about the role of the husband in the prognosis and treatment of PPD

Experimental design

6) 99 A descriptive correlation cross-sectional design—its descriptive cross-sectional design 

7)97 Materials & Methods—not material, its subject 

8) 154 gender of children, gender of children, and satisfaction with the gender of the baby------ should be sex of children.

9) 154 & 160 close-ended questions ----- closed-ended questions

10) Table 1: ------50 years --correct 

11) Table 1: many variable choices start with small letters

12) The p value should be italic. 

13) add the relationship between demographic determinants and the total knowledge score, and discuss

14) add the relationship between demographic determinants and total attitude scores.to be more insightfully

15) add the questionnaire in the supplementary material in Arabic

16) what is the write answer for each of the knowledge questions

17) you should split no --- from don't know
as no (as they differ

Validity of the findings

What about the validity process?

·

Basic reporting

English is fluent. However, there are some sentences that requires clearer explanation. Please see my comments in the attached PDF.

Experimental design

Good. There are some comments on recruitment and sampling method. Please see the attached PDF.

Validity of the findings

Please see attached PDF.

---

## Round 0.2 · Major Revisions

Even the authors provide efforts to improve the quality of the manuscript; the latter still lacks a certain rigor and should be revised.

The main concern related to those who did not hear of this condition is still existing. The authors just deleted the coresponding sentences without providing the appropriate analysis.

The manuscript still lacks certain rigor related to the way of writing and the English language. The flow of the ideas in the introduction is still not adequate. The first three sentences are not referenced and not correlated, for example.
In addition, even though the introduction is very long, the hypothesis is not well defined since a study that was conducted in the same country was not cited as mentioned by reviewer 5 "Alkhawaja, A., Alwusaibie, F., Almarzooq, A., Alwosaifer, A., & Eltwansy, M. S. (2023). Knowledge and Attitude of Husbands Towards Postpartum Depression in Dammam, Khobar and Qatif, Saudi Arabia. Volume 46, Issue 11, ISSN: 03875547.”

All studies conducted in the country should be mentioned.

You should add the name of the country and the title.

You should provide more "quantitative" results in the abstract.

Try to improve the quality of the tables and reduce their length (especially Table 4).

Change "minimal" to "low knowledge.".

Reviewer 4 ·

Basic reporting

The basic reporting of this manuscript is good and does not hinder readers from understanding the study.

Experimental design

The experimental design was explained thoroughly and also reproducible.

Validity of the findings

The presented data is plausible and credible.

Additional comments

A review of the manuscript entitled “Husbands’ knowledge and attitudes regarding postpartum depression”

- Authors are suggested to proofread the manuscript after addressing all comments to avoid any typological, grammatical, and lingual mistakes and errors. For example, on line 26 “…were ages 50 or older Almost…”; line 142 “and Single males”; line 408.
- Whether it's uppercase or lowercase, the manuscript should consistently use the same alphabet to display the p-value. The current version still differs in terms of consistency. Sometimes the authors report it using lowercase, then other times using uppercase. Please revise.
- (lines 46-47) This sentence: “The incidence of PPD has increased among Arab women and is expected to affect at least one in five mothers.” also shares the same idea with a similar study by Harahap et al. https://pubmed.ncbi.nlm.nih.gov/38798854/ Kindly include this reference to make the sentence stronger.
- (line 131) I suggest that the authors include their hypothesis right after the research question to show whether it is rejected or not after the results are obtained.
- (lines 49-52) The authors describe symptoms related to PPD in the following sentence well; however, I encourage authors to add one more problem named “feeding problems” which also contributes to PPD, to enrich the sentence. “Tearfulness, anxiety, emotional stress, irritability, sleep disorders, memory problems, guilty feelings, loss of appetite, feeding problems, and suicidal thoughts, in addition to feelings of weakness and inability to deal with the baby, characterize PPD.” Please cite https://doi.org/10.52225/narrax.v2i3.163
- Please revise the sentence on lines 515-516 by specifying that this result demonstrates a positive relationship between a high level of knowledge and attitude among Saudi husbands.
- The sentence on lines 517-518 is repetitive, please omit it.

Reviewer 5 ·

Basic reporting

clear and literature references sufficient,

Experimental design

the research question well defined

Validity of the findings

The manuscript does not include any advanced statistical analysis, which limits the depth of interpretation and the robustness of the findings.
Additionally, a similar study has already been conducted in the same country but in a different region or city, using the same data collection tools. The authors need to provide a meaningful rationale for conducting the same study

Additional comments

need more advanced statistical analysis

---

## Round 0.3 · Major Revisions

The authors should take into consideration the comments provided by reviewer 4.
They should revise their manuscript rigorously for the spelling errors mentioned by the same reviewer.

**Language Note:** The review process has identified that the English language must be improved. PeerJ can provide language editing services - please contact us at [email protected] for pricing (be sure to provide your manuscript number and title). Alternatively, you should make your own arrangements to improve the language quality and provide details in your response letter. – PeerJ Staff

Reviewer 4 ·

Basic reporting

- Authors need to proofread their manuscript to ensure that errors in the writing process do not exist. For example, an extra space, which is not needed before the percentage symbol on line 60 (…in the sample was 13.6 %...), an unnecessary bracket on line 63, “I do not’t know” on line 224, line 332.

Experimental design

- I found inconsistency on lines 224-231. The authors said that the total knowledge scores ranged from 0 to 30. However, the next classification showed 33 as the highest score. This is not in accordance with the previous statement and should be revised.

Validity of the findings

No comment.

Additional comments

- The sentence on line 73, “Cultural factors that can develop to PPD giving birth to a female”, is incomplete. Please revise it for clarity.

- This statement, “Most postpartum women develop PPD while staying at home”, needs more elaboration as it might lead to ambiguity. In general, everybody stays at home. Do you mean those who do not have other job-related activities outside their homes? Or those who are housewives?

- This sentence on line 185 should be written in a gender-neutral tone: “The researcher introduced herself and clearly explained the study’s aims.”
- (line 241-243) what is the mean score used to determine whether attitudes were categorized as positive or negative?

- (line 246) this sentence is unclear, “The Arabic version was revised by five experts in the field of nursing studies at KAU”, as the authors did not specify what it is in the Arabic version that they mean here. Were they the questionnaires?

- (line 261) the purpose for doing this, “Negatively worded items were reverse coded before the analysis.”, should be explicitly explained in your methods.

- The introduction covers sufficient contextual background to understand this study, however, I find it somewhat abrupt to start directly with PPD in the beginning of this section. Therefore, to address this, I suggest that the authors begin with broader context, such as what depression itself is, before jumping further into PPD. For example, “A major medical disorder connected with bad emotions and a decline in quality of life is depression. With the greatest rates recorded in the Middle East and South Asia, the global frequency of depression reached 33%.” These statements can be cited from https://pmc.ncbi.nlm.nih.gov/articles/PMC10914089/

Reviewer 5 ·

Basic reporting

clear and literature references are sufficient

Experimental design

research question well defined

Validity of the findings

all needed correction was done

---

## Round 0.4 · accepted · Accept

The authors have addressed all of the reviewers comments. With these amendments, the manuscript is ready for publication.

Reviewer 4 ·

Basic reporting

no comment

Experimental design

no comment

Validity of the findings

no comment